# Epigenetic Modulation of Inflammatory Pathways in Myometrial Stem Cells and Risk of Uterine Fibroids

**DOI:** 10.3390/ijms241411641

**Published:** 2023-07-19

**Authors:** Qiwei Yang, Mohamed Ali, Lindsey S. Treviño, Aymara Mas, Nahed Ismail, Ayman Al-Hendy

**Affiliations:** 1Department of Obstetrics and Gynecology, University of Chicago, Chicago, IL 60637, USA; mohamed.ali@bsd.uchicago.edu; 2Division of Health Equities, Department of Population Sciences, City of Hope, Duarte, CA 91010, USA; ltrevino@coh.org; 3INCLIVA Health Research Institute Avda, Menéndez Pelayo 4, 46010 Valencia, Spain; amas@fundacioncarlossimon.com; 4Department of Pathology, College of Medicine, University of Illinois at Chicago, Chicago, IL 60612, USA; ismail7@uic.edu

**Keywords:** uterine fibroids, myometrial stem cells, diethylstilbestrol, developmental reprogramming, inflammatory responsive genes

## Abstract

The period during which tissue and organ development occurs is particularly vulnerable to the influence of environmental exposures. However, the specific mechanisms through which biological pathways are disrupted in response to developmental insults, consequently elevating the risk of hormone-dependent diseases, such as uterine fibroids (UFs), remain poorly understood. Here, we show that developmental exposure to the endocrine-disrupting chemical (EDC), diethylstilbestrol (DES), activates the inflammatory pathways in myometrial stem cells (MMSCs), which are the origin of UFs. Significantly, the secretome of reprogrammed MMSCs enhances the expression of critical inflammation-related genes in differentiated myometrial cells through the paracrine mechanism, which amplifies pro-inflammatory and immune suppression signaling in the myometrium. The expression of reprogrammed inflammatory responsive genes (IRGs) is driven by activated mixed-lineage leukemia protein-1 (MLL1) in MMSCs. The deactivation of MLL reverses the reprogramming of IRG expression. In addition, the inhibition of histone deacetylases (HDACs) also reversed the reprogrammed IRG expression induced by EDC exposure. This work identifies the epigenetic mechanisms of MLL1/HDAC-mediated MMSC reprogramming, and EDC exposure epigenetically targets MMSCs and imparts an IRG expression pattern, which may result in a “hyper-inflammatory phenotype” and an increased hormone-dependent risk of UFs later in life.

## 1. Introduction

Uterine fibroids (UFs) are benign tumors of the uterus arising from the myometrium, with an estimated prevalence of ~70% among women of reproductive age [1,2,3]. Despite their widespread occurrence and significant impact on public health, the options for treatment are still quite restricted, with a hysterectomy the most common one [1,4]. Multiple risk factors that contribute to the development of UFs, including the harmful environmental exposure to substances such as endocrine-disrupting chemicals (EDCs), have been identified. Compelling evidence suggests that exposure to hormones during development may be associated with a predisposition of the myometrium to UF development [5,6]. Among them, EDCs, such as diethylstilbestrol (DES), phthalates, and the soy phytoestrogen, genistein, have contributed to enhancing the occurrence, number, and size of UFs in animal models [7,8,9,10]. Furthermore, epidemiologic studies on humans have found an association between an increased risk of early UF diagnosis and in-utero DES exposure, as well as the consumption of soy-based formulas during infancy [10,11,12].

Given that UFs are characterized as monoclonal tumors, it is plausible that the dysregulation of committed cells that acquire stem-like characteristics could be accountable for the development of this non-malignant condition. UFs are monoclonal tumors that originate in the myometrium. An increasing body of evidence supports the hypothesis that UFs originate from stem cells in the myometrium. Previous studies have been performed by our group and others to identify tumor-initiating cells in UFs [13,14,15,16]. Additionally, we demonstrated that early-life exposure to EDCs decreased the DNA repair capacity in myometrial stem cells (MMSCs) from Eker rats [17,18], highlighting the important role of DNA damage repair pathways in UF pathogenesis. Besides the DNA repair pathway, it has also been reported that early-life exposure to EDCs creates a prolonged inflammation status, and inflammation plays an important role in the initiation and development of many types of diseases, including tumorigenesis [1]. These links are consistent with the perinatal programming of intestinal inflammatory disorders [19]. However, how the impact of early-life exposure to adverse factors on the inflammatory pathway in MMSCs remains unclear. In the present study, we first assessed whether developmental exposure to EDCs during sensitive periods of uterine development reprograms the inflammatory pathways in MMSCs. We then determined how developmental EDC exposure triggers alterations in the epigenome related to the inflammatory pathway. Overall, our findings reveal that early-life exposure to EDCs increases the risk of UF development by reprogramming the epigenome toward the activation of inflammatory pathways.

## 2. Results

### 2.1. Identification of Distinct Inflammatory Responsive Genes in DES- vs. VEH-Exposed MMSCs of Eker Rats

To identify the differences in gene expression that could potentially contribute to the development of UFs within the context of the inflammation pathway, we used the Eker rat model. This particular animal model has been widely regarded for investigating the interactions between genes and the environment in relation to UF development so far [20]. We subjected newborn Eker rats to either DES or VEH between postnatal days 10 and 12. When the rats reached 5 months of age, we obtained myometrial tissues from them, and isolated the rMMSCs using Stro-1 and CD44 surface markers (see the experimental design in Figure 1A). Through Over-Representation Analysis (ORA) using hallmark gene sets, we identified the enrichment of the epithelial–mesenchymal transition pathway, along with several pathways associated with inflammation. These include NFkB-TNFa, inflammatory response, and IL2-STAT5 signaling (Figure 1B). Furthermore, when we performed the Ingenuity Pathway Analysis (IPA) of differentially expressed genes (DEGs), we found that TGF-β, TNF, VEGF, and PDGF act as upstream regulators. These molecules are particularly significant as they are widely recognized factors involved in inflammatory events, suggesting their potential to modulate inflammatory pathways in the DES-MMSCs. Next, we compared the differential expression of IRGs between the VEH-MMSCs and DES-MMSCs. Out of 111 genes associated with a hallmark inflammatory response that can be detected (Figure 1B), we observed that 34 genes were upregulated, and 18 genes were downregulated in the DES-MMSCs compared to those in the VEH-MMSCs (Figure 1B). Figure 1C and D list the top 18 upregulated and downregulated IRGs, respectively, in the DES-MMSCs compared to those in the VEH-MMSCs.

To validate the IRGs expression in the DES-MMSCs and VEH-MMSCs from RNA-seq data, *ccl2*, *lpar1*, *pdpn*, and *ccl7* were selected based on their important roles. The latter is a potent inflammatory chemokine as it can bind to numerous C-C motif chemokine receptors (CCR), including CCR1, CCR2, CCR3, and CCR5, on many types of cells. In addition, CCL7 acts as a chemoattractant for multiple leukocytes, therefore mediating the immune response. As shown in Appendix A, the expression levels of *ccl7*, *ccl2*, *lpar1*, and *pdpn* are significantly upregulated in the DES-MMSCs compared to those in the VEH-MMSCs, which is consistent with the RNA-seq data. In addition, our RNA-seq data demonstrated that several other genes related to inflammatory promoters or receptors, including *Il7*, *Il16*, *Il33*, *Il34*, *Il17b*, *Il4r*, and *Il6r*, were more upregulated in the DES-MMSCs compared to those in the VEH-MMSCs, indicating that developmental exposure to EDCs activates the pro-inflammatory pathway.

### 2.2. DES Exposure Reprograms the Inflammatory Responsive Genes in MMSCs

To determine the epigenetic mechanism underlying the altered expression pattern in response to early-life EDC exposure, we performed global ChIP-seq using the active epigenetic mark, H3K4me3. We then integrated the gene expression data from RNA-seq with the H3K4me3 enrichment data from ChIP-seq in the DES- and VEH-MMSCs. As shown in Figure 2A, out of the 16 up-regulated IRGs analyzed, 14 IRGs showed an increase in the H3K4me3 levels, while 2 IRGs displayed a decrease in the H3K4me3 levels. All of the six down-regulated IRGs analyzed showed a decrease in the H3K4me3 levels. Fisher’s exact test showed a significant correlation between IRG expression and the H3K4me3 status (*p* < 0.001). Moreover, H3K4me3 reprogrammed 21 IRGs without altering the gene expression levels (Figure 2A). Figure 2B lists the IRGs, demonstrating a correlation between the H3K4me3 status and RNA expression levels. Figure 2C lists the representative reprogrammed IRGs, showing histograms created using the Integrative Genomics Viewer, further demonstrating H3K4me3 occupancy at the promoter regions of these pro-inflammatory genes, including *Ccl7*, *Ccl2*, *Ereg*, *Cd40*, *Pcdh7*, and *Ptger2*. The products encoded by these IRGs play an essential role in many cellular functions and serve as pro-inflammatory cytokines, growth factors, and transcription factors involved in immune responses and tumorigenesis (Figure 2D).

### 2.3. Reprogrammed MMSCs Activates Inflammatory Pathway in Myometrium

To determine whether reprogrammed MMSCs influenced the characteristics of differentiated myometrial cells (DMCs) in the myometrium, we isolated adult rat DMCs from normal myometria and exposed them to the conditioned medium (CM) from the VEH-MMSCs or DES-MMSCs. We selected several important genes for the measurement in response to exposure to the CM based on their critical roles in pro-inflammatory pathways such as the inflammasome pathway. As shown in Figure 3, secreted factors (secretome) from the DES-MMSCs significantly increased the expression of several critical pro-inflammatory genes, including *ccl2*, *ccl7*, *pdpn*, and *lpar1* (Figure 3, left panel), as well as *Il-1a*, *Il-1b*, *Il-6*, and *Tnf-a* (Figure 3. right panel) in the DMCs, suggesting that DES exposure may trigger the inflammasome pathway. The RNA-seq data revealed that DES exposure disrupted the normal expression of *casp1* and *nlrp3*, which are the genes associated with the inflammasome family. Consequently, DES may have the potential to intensify pro-inflammatory and immune-suppressive signaling within the myometrium through the involvement of MMSCs.

To determine if IRG reprogramming via early-life exposure to DES is MMSC-specific, we compared the expression of IRGs between the MMSCs and differential myometrial cells (DMCs, Stro-1^−^/CD44^−^) from the myometria of Eker rats developmentally exposed to DES. As shown in Figure 4A, out of six IRGs (*Itgb3*, *Kcna3*, *Tlr2*, *Sgms2*, *Pik3R5*, and *Kif1b*) with enriched H3K4me3, three genes (*Kcna3*, *Tlr2*, and *Sgms2*) were significantly altered. For five genes with the reduction of H3K4me3, three of the genes, including *Irf7*, *Itga5* and *Tnfsf9*, exhibited a significant difference between the MMSCs and DMCs (Figure 4B). Notably, out of eleven IRGs examined, five genes (*Itgb3*, *Kcna3*, *Tlr2m Irf7*, and *Itga5*) showed either an opposite expression pattern, or one showed a change, and the other one showed no changes between the DES-MMSCs and Stro-1/CD44-negative cells while using VEH-MMSCs as a reference. In addition, the expression of *Sgms2* and *Tnfsf9* showed a significant difference between the MMSCs and DMCs, but the changes were the same if VEH-MMSCs were used as a reference. Therefore, developmental exposure to DES reprogrammed some IRGs in the MMSCs more specifically compared to that of the DMCs (Figure 4C).

### 2.4. EDC Exposure Reprograms IRGs via H3K4me3 Mechanism

Mixed-lineage Leukemia1 (MLL1) is a well-studied mammalian homolog of the yeast protein Set1 that is cleaved by Taspase1 (Tasp1) to generate a mature, heterodimerized MLL1 N320/C180 methylate histone H3 at K4, leading to an activated H3K4me3 marker [21]. To determine the mechanism underlying increased expression of IRGs in response to early-life exposure to DES, we knocked down *Tasp1* using a lentiviral shRNA approach. As shown in Figure 5A, *Tasp1* shRNAs efficiently reduced the expression of *Tasp1*. In addition, among the four genes tested. *Tasp1* shRNAs led to a significant decrease in the RNA expressions of *ccl2*, *ccl7*, and *lpar1* (Figure 5B), suggesting that the deactivation of MLL1 via *Tasp1* knockdown decreased the number of H3K4me3-reprgrammed IRGs in the DES-MMSCs. 

### 2.5. Inhibition of HDACs Reversed the DES-Induced Aberrant Expression of IRGs in MMSCs

Previously conducted research has put forth the idea that HDACs can interact with MLL1 [22]. HDAC activity is linked to a variety of diseases, and intervention with HDAC inhibitors suppresses the phenotype of diseases, including UFs [23]. In this regard, we determined if the inhibition of HDACs could reverse the DES exposure-induced upregulation of IRGs. Therefore, DES-MMSCs were treated with an HDAC inhibitor (HDACi VIII) for 1 and 2 days at concentrations of 2.5 mg/mL and 5 mg/mL, respectively. As shown in Figure 6, the HDAC inhibitor dose-dependently decreased the expression of the IRGs, including *ccl7*, *ccl2*, *lpar1*, and *pdpn*, at day 1 and day 2, in the DES-MMSCs, suggesting that HDAC inhibition reverses the reprogrammed IRGs induced by developmental DES exposure.

## 3. Discussion

Early-life exposure to a variety of insults during sensitive windows of development can reprogram normal physiological responses and alter disease susceptibility later in life [24,25,26]. During this process, inflammation triggered by various types of adverse exposure plays an important role in the initiation and development of many types of diseases, including tumorigenesis [27]. Although remarkable advances have been achieved in determining the role of inflammation in the pathogenesis of diseases, little information is known about the origin and initial steps of UF development related to inflammation. Here, we used Eker rats that carry a germ-line mutation in the tuberous sclerosis complex 2 (*tsc2*) tumor suppressor gene and are susceptible to developing UFs. Through the isolation of myometrial Stro1^+^/CD44^+^ stem cells, which are involved in the formation of UFs, we addressed the inquiry regarding the impact of developmental exposure to endocrine-disrupting chemicals (EDCs) on the cellular origin of UFs and its association with an elevated risk through an inflammatory pathway. Our findings demonstrate that developmental exposure to EDCs causes the reprogramming of the MMSC/progenitor cell population, leading to an activated inflammatory signaling pathway during the early stages of UF development. This pro-fibroid stage ultimately contributes to the elevated risk of UF formation later in life. 

Previous studies have been conducted to characterize the role of inflammation in UFs [28,29]. These studies have demonstrated that numerous pro-inflammatory mediators that trigger or enhance specific aspects of inflammation are more upregulated in UF tumors than adjacent myometrium tissues. In addition, the levels of tumor necrosis factor TNF-α, a cell-signaling protein involved in systemic inflammation, are elevated in women with clinically symptomatic UFs [30]. A recent study demonstrated that higher numbers of macrophages are present inside and close to UFs as compared to those in the more distant myometrium [31]. Notably, several key pro-inflammatory mediators, such as TGF-β are overexpressed in UFs [32,33], a potent chemoattractant factor for macrophages. At the progenitor cell level, UF progenitor cells secrete significantly higher levels of cytokines related to chronic inflammation, while producing significantly lower amounts of cytokines associated with acute inflammation [34]. These studies suggest that inflammation is linked to the UF phenotype. In the present study, we want to address the important question of how UF pathogenesis is initiated. By using a unique animal model in combined with progenitor/stem cell omics studies, we demonstrated for the first time that MMSCs from at-risk myometria exposed to EDCs have a more activated inflammatory pathway compared to that of VEH-exposed MMSCs, which may partly explain why DES exposure increases the risk of UF development.

We also identified the altered biological pathway in MMSCs in response to early-life exposure to EDCs using omics analyses. Our proof of concept using RNA-seq analyses with the Eker rat animal model demonstrates that early-life exposure to EDCs results in the enrichment of gene sets associated with pro-inflammatory pathways. Additionally, IPA analysis revealed that several factors, such as TGF-β and TNF, act as upstream regulators influencing key pathways. These findings suggest that EDCs play a role in regulating inflammatory signaling in MMSCs.

Importantly, we demonstrate that EDC-MMSCs are capable of impairing differentiated cells via enhancing inflammatory pathways by increasing the key pro-inflammatory gene expression. Given the fact that the percentage of progenitor cells is much lower compared to that of differentiated cells, one may consider that reprogrammed MMSCs, due to EDC exposure, may amplify their inflammatory response via affecting the surrounding DMCs, leading to a pro-inflammatory milieu, which can act in a paracrine manner on the infiltrating immune cells. The accumulated evidence demonstrates that chronic inflammation-induced DNA damage subsequently results in many diseases, including tumorigenesis. The failure to repair DNA damage or to control the inflammatory responses has the potential to increase the risk of tumorigenesis [35]. The DNA repair pathway is essential in preventing genome instability and DNA mutation. But inflammation can cause DNA damage by interfering with the DNA repair mechanism [36]. Our previous studies showed that developmental exposure to EDCs caused DNA damage by decreasing the DNA repair capacity [17,18] in MMSCs. The results of this study indicate a potential interplay between DNA repair mechanisms and inflammation. This connection may be associated with an increased susceptibility to the development of UF pathogenesis.

To identify a molecular mechanism by which developmental exposure to DES precedes the development of UFs, we conducted ChIP-seq studies using MMSCs exposed to DES during a critical period of uterine development. We found that developmental exposure to DES during uterine development reprograms the inflammation responsiveness of targets in MMSCs. The selective developmental programming of enriched IRGs was observed in the MMSCs in response to DES exposure before tumor development occurred. MMSCs were isolated from 5-month-old animals, representing the early adulthood phase before UF development. The integrated multi-omics analysis of RNA-seq and ChIP-seq demonstrated a significant correlation between the aberrant expression of IRGs and H3K4me3 enrichments, indicating that EDC exposure may disrupt the MMSC epigenome, leading to an altered IRG expression.

Mixed-lineage Leukemia1 (MLL1) is an initial 500-kDa protein that is cleaved by Taspase1 (Tasp1) to generate a mature MLL1 N320/C180 heterodimer [21]. MLL1 cleavage/activation can methylate histone H3 at K4, resulting in the activation-associated H3K4me3 modification. Notably, MLL1 has been shown to be involved in the inflammation process and regulate the TNFa-stimulated activation of genes downstream of NF-kB [37]. In this regard, we performed a loss-of-function experiment to knockdown *Tasp1* so that we can block the production of the mature form of MLL1. Then, we examined the expression levels of reprogrammed IRGs, which play a critical role in the modulation of proinflammatory progression. Our study demonstrated that MLL1 is required for the regulation of IRG expression. Among the four IRGs we examined, all exhibited a reduced expression after *Tasp1* knockdown. Notably, a previous study showed that MLL1 is involved in the early-life BPA-induced risk of prostate cancer [21]. Therefore, developmental EDC exposure may increase the risk of tumorigenesis by stimulating the proinflammatory process through MLL1 activation.

In addition to EDC-induced DNA repair dysfunction in MMSCs [17,18,38], and the reported interplay between inflammation and DNA repair in other diseases [39,40,41,42], it is noted that the reprogrammed IRGs identified in this study have multiple functions. For instance, Ccl7, as a potent inflammatory mediator, triggers multiple pathways via binding to several functional CCRs [43]. CCL7 can recruit immune cells that express associated receptors to further amply inflammatory processes and contribute to disease progression. In this study, we identified the increased expression of *ccl7* associated with elevated H3K4me3 in DES-MMSCs, suggesting that CCL7 may enable MMSCs to over-stimulate inflammatory responses and allow MMSCs to exhibit an exaggerated effect after a second insult later in life. Moreover, some of the reprogrammed IRGs are MMSC-specific when the comparison of IRGs expression between DES-MMSCs and DES-DMCs was performed. Therefore, further investigation of the molecular mechanism underlying the cell-type-specific gene regulation in response to EDC exposure is needed.

The formation and activation of inflammasome complexes is considered an important step in the inflammatory form of cell death [44]. The NLRP3 inflammasome complex modulates innate immune activity and inflammation. This inflammasome complex triggers the activation of caspase 1, a cysteine protease, which produces active pro-inflammatory interleukins [45,46,47]. In this study, we demonstrated that NLRP3 is upregulated in DES-MMSCs, suggesting that the NLRP3 inflammasome signaling pathway may be involved in an increased risk of MMSCs. Additionally, we observed reprogrammed MMSCs exhibit an increase in the expression of key interleukins, such as IL-1a, Il-1b, and IL-6, among others, in neighboring DMCs through paracrine mechanisms. A deep insight into characterizing the status and role of the NLRP3 inflammasome in DMCs that regulate the release of pro-inflammatory cytokines will help to us to better understand the interaction between the MMSCs and surrounding DMCs contributing to the pathogenesis of UFs.

It is reported that the interaction between MLL1 and HDACs can regulate gene expression [22]. In this regard, we determined if targeting HDACs could reverse the aberrant expression of IRGs induced by EDC exposure. Similarly, HDAC inhibition showed a dose-dependent suppression of IRG expression. Our results highlight the role of histone modifications as a regulatory platform to orchestrate the expression of inflammation-related genes by EDC exposure. Notably, we identified 21 reprogrammed IRGs without changes in the RNA expression at the base level between EDC-MMSCs and VEH-MMSCs. These developmentally reprogrammed genes were not a consequence of altered expression but may be primed to have an exaggerated response to environmental exposure after a second hit. Wang et al. reported that several BPA-reprogrammed genes with increased H3K4me3 levels were expressed at the same levels as that of the vehicle control in the adult prostates of rats neonatally exposed to BPA. However, 6 h after the hormone treatment (testosterone plus estradiol), 7/9 of these genes exhibited a significantly increased expression in response to the hormone in d70 rats that had been neonatally exposed to BPA in later life relative to that of the BPA-treated rats that did not receive a hormone treatment. In contrast, only 1/9 of the genes became significantly elevated in d70 rats that had been neonatally exposed to the vehicle compared to that of the vehicle-treated rats that did not receive the hormone [21]. Therefore, our study supports the idea that early-life environmental exposure can reprogram the epigenome, increasing the risk of diseases in adulthood [25,48,49,50].

In summary, our findings provide compelling evidence that MMSCs are the direct epigenetic targets of xenoestrogen (DES) actions and illustrate the strength of epigenomic alteration in revealing key gene sets that modulate the pro-inflammatory pathways of MMSCs. Early-life exposure to EDCs (e.g., DES) reprograms the patterns of expression of IRGs in a way that is mediated by histone in MMSCs. TASP1 and HDAC activities are required for the altered expression of IRGs in response to early-life exposure to EDCs. These changes in gene expression alter the characteristic features of MMSCs, resulting in an increased risk of hormone-dependent uterine-related diseases, such as UFs, by amplifying the immune response in the surrounding myometrium cells (Figure 7). Notably, due to the different characteristics of each EDC and environmentally relevant EDC mixtures, developing an effective and conventional treatment strategy for EDCs will be challenged. Therefore, a deep insight into EDC-induced reproductive dysfunction will be helpful to develop new therapies for the treatment, or preferably the prevention, of UF pathogenesis while limiting collateral damage.

## 4. Materials and Methods

### 4.1. Postnatal Exposure to the Endocrine Disruptor DES

Ethical approval for all experiments involving female Eker rats (Long Evans; Tsc-2Ek/+) was obtained from the Augusta University Animal Care Committee and Baylor College of Medicine. The estrous stage of the animals was determined through the histological examination of the vagina and analysis of serum levels of ovarian hormones in all instances [20]. To investigate the interplay between early-life environmental exposure and the reprogramming of myometrial stem cells, neonatal female Eker rats sourced from an in-house colony were subjected to subcutaneous injections. Each rat received a daily injection of 10 µg of DES (Sigma, St. Louis, MO, USA) (n = 5), or 50 µL of sesame seed oil (vehicle, VEH, n = 5) on days 10, 11, and 12 after birth (PND10-12) following a previously established protocol [20]. The animals were raised until they reached 5 months of age, at which time they were euthanized, and myometrial Stro1^+^/CD44^+^ stem cells with DES or VEH exposure were isolated from their tissues. Simultaneously, we also collected Stro-1^−^/CD44^−^ cells from DES-exposed myometria.

### 4.2. Isolation and Characterization of Myometrial Stro-1^+^/CD44^+^ Cells from Eker Rats Exposed to DES and VEH

Myometrial Stro-1^+^/CD44^+^ cells from Eker rats were isolated as previously described [51]. Briefly, the uterine tissues from Eker rats were collected and rinsed in a wash buffer solution (Life Technologies, Grand Island, NY, USA). The myometrial layer was isolated by removing the endometrium and serosa with a sterile scalpel. Subsequently, freshly isolated myometrial cell suspensions were stained with antibodies to the cell surface proteins Stro-1 and CD44 (BD Biosciences, San Jose, CA, USA) and sorted via flow cytometry. The isolated Stro-1^+^/CD44^+^ cells were grown in collagen-coated dishes in Smooth Muscle Growth Medium (SmGM, Lonza, Walkersville, MD, USA) supplemented with 5% fetal bovine serum, 0.1% insulin, 0.2% hFGF-B, 0.1% GA-10000, and 0.1% hEGF (Lonza). The cells were maintained at 37 °C in an incubator in a humidified atmosphere at 2% hypoxia. The characterization of rMMSCs was performed as previously described [51].

### 4.3. Whole-Transcriptome RNA Sequencing (RNA-seq) and Quantitative Real-Time RT-PCR (RT-qPCR)

To identify genes that undergo developmental reprogramming and are influenced by DES, whole-transcriptome sequencing and data analysis were conducted. Myometrial Stro-1^+^/CD44^+^ stem cells obtained from the uteri of 5-month-old animals that had been exposed to DES on PND10-12 were used for this purpose. The RNeasy RNA isolation kit (Qiagen, Valencia, CA, USA) was used to isolate RNA for RNA-seq analysis. Prior to cDNA synthesis, RNA samples were treated with DNase I, and the purity and quality of RNA were checked using a Bioanalyzer. cDNA libraries were constructed using SPRI-works Fragment Library System I (Beckman Coulter, Brea, CA, USA), and were then PCR-enriched and purified. For cluster generation, 10 pM DNA was loaded into the paired-end flow cell using the cBOT system, and then loaded on the HiSeq2500 platform (Illumina Inc., San Diego, CA, USA) to generate single 36 bp sequence reads. Sequence reads were aligned to the rat reference genome rn6 using Hisat2, and aligned read counts were analyzed using EdgeR for differential gene expression.

For qPCR, the same RNAs used for RNA-seq were subjected to reverse transcription into first-strand cDNA using RNA to cDNA EcoDry Premix (Takara Bio USA, San Jose, CA, USA) following standard procedures. All qPCR assays were performed in the 96-well format, and each sample was run in triplicate to ensure accuracy and reproducibility. The real-time fluorescence detection of PCR products was performed using the following thermocycling conditions: 1 cycle of 95 °C for 2 min followed by 40 cycles of 95 °C for 5 s and 60 °C for 30 s. The sequences of the primers used in this study are listed in Appendix A. For gene expression,18S ribosomal RNA was used as an endogenous control. For data analysis, the comparative method (∆∆Ct) was used to calculate the relative quantities of nucleic acid sequence.

### 4.4. Chromatin Immunoprecipitation Sequencing (ChIP-seq)

ChIP-seq was performed as described previously [21]. A chromatin immunoprecipitation kit (EZ-Magna ChIP™, Millipore, Billerica, MA, USA) was used according to the manufacturer’s protocol to prepare chromatin from myometrial Stro-1^+^/CD44^+^ stem cells. These cells were then treated with 1.5% formaldehyde to crosslink proteins to DNA. Subsequently, they were then incubated with 125 mM glycine and rinsed with cold PBS. After centrifugation, cells were resuspended in cell lysis buffer (PBS containing 0.5 mM EDTA and 0.05% Triton X-100) and collected again through centrifugation. Each cell pellet was resuspended in nuclear lysis buffer (50 mM Tris-HCl, pH 8.1, 10 mM EDTA, and 1% SDS). Chromatin preparations were sonicated in a Bioruptor (Diagenode, Denville, NJ, USA) to obtain fragments comprising 100–500 bp in length. ChIP was performed according to the instructions provided with the Magna ChIP kit using the antibody against H3K4me3 (Active Motif, Carlsbad, CA, USA).

ChIP-seq data analysis was performed as described previously [21]. Briefly, sequence reads from the ChIP experiment were de-multiplexed and converted to FASTQ files. The resulting FASTQ files were first aligned to the same reference rat genome as that used in the RNA-seq data analysis using bwa [52] to create BAM files with alignment information. MACS (Model-based Analysis of ChIP-Seq) [53] was then performed to identify the peaks in different histone-marked IP samples relative to the corresponding input. The aligned BAM files from the previous step were used in the MACS peak calling process with the lower and upper limits for fold enrichment parameters set to 4 and 30, respectively.

### 4.5. Knockdown of Tasp1 in MMSCs from Eker Rats

To determine methyltransferase specificity for changes in H3K4me3 in response to DES exposure, the transient knockdown of Taspase1 (*Tasp1*) expression was performed using viral particles from a rat shRNA construct (OriGene, Rockville, MD, USA). DES-MMSCs were incubated with the virus multiplicity of infection (MOI, 10 PFU/mL) overnight in 6-well plates, the medium was then changed, and cells were grown for 5 days. RNAs were prepared and used to measure the RNA levels in MMSCs with *Tasp1* rat shRNA lentiviral particles or scrambled controls.

### 4.6. HDAC Inhibition in MMSCs from Eker Rats

To determine the effect of HDAC inhibition on the reversion of reprogrammed IRGs, DES-MMSCs were grown in the presence or absence of HDAC inhibitor (HDACi VIII) at a concentration of 2.5 mg/mL or 5 mg/mL for 1 or 2 days. The cell pellets were then collected and subjected to RNA isolation, cDNA synthesis, and qPCR analysis.

### 4.7. The Effect of MMSC Secretome on the Expression of Proinflammatory Factors in DMCs

Conditioned medium from MMSCs was prepared as previously described [54]. Differentiated myometrial cells (DMCs) were isolated from rat adult myometrium using our previous methods [55]. To determine the paracrine effect of reprogrammed MMSCs on DMCs, the DMCs were grown in the presence of conditioned medium collected from either EDC-MMSCs or VEH-MMSCs for two days, before the performance of assays.

### 4.8. Statistical Analysis

For statistical analysis, flow cytometric data were analyzed using FlowJo 8.7.3 software. The results were expressed as mean ± SEM using GraphPad Prism version 5.00 for Windows (GraphPad Software, San Diego, CA, USA). To determine the significant difference, comparisons were performed using a two-tailed t-test for groups or a two- or one-way ANOVA test for three or more groups, with a *p*-value ≤ 0.05 being considered statistically significant. In addition, Fisher’s exact test was used to examine the correlation between RNA expression and H3K4me3 enrichment of reprogrammed inflammatory responsive genes (IRGs).

### 4.9. Bioinformatics Analysis

Top biological functions and canonical pathways associated with the differentially expressed mRNA data set were identified using Ingenuity Pathway Analysis (IPA) (Qiagen). A *z*-score was calculated to infer the activation states of implicated biological processes. Over-Representation Analysis (ORA) was used to determine the over-represented biological processes in our experimentally derived gene list.

## 5. Conclusions

We present compelling evidence demonstrating that developmental exposure to EDCs targets the inflammatory pathways in MMSCs, the cell source for UFs. The reprogrammed IRGs may result in a “hyper-inflammatory phenotype” and an increased hormone-dependent risk of UFs with a second insulted hit later in life. Targeting MLL1 and HDAC epigenetic regulators can reverse notable changes in the expression of IRGs in the MMSCs from Eker rats exposed to EDCs. We hope this preclinical study will help researchers to understand better the impact of developmental exposure insults on human health.

## Figures and Tables

**Figure 1 ijms-24-11641-f001:**
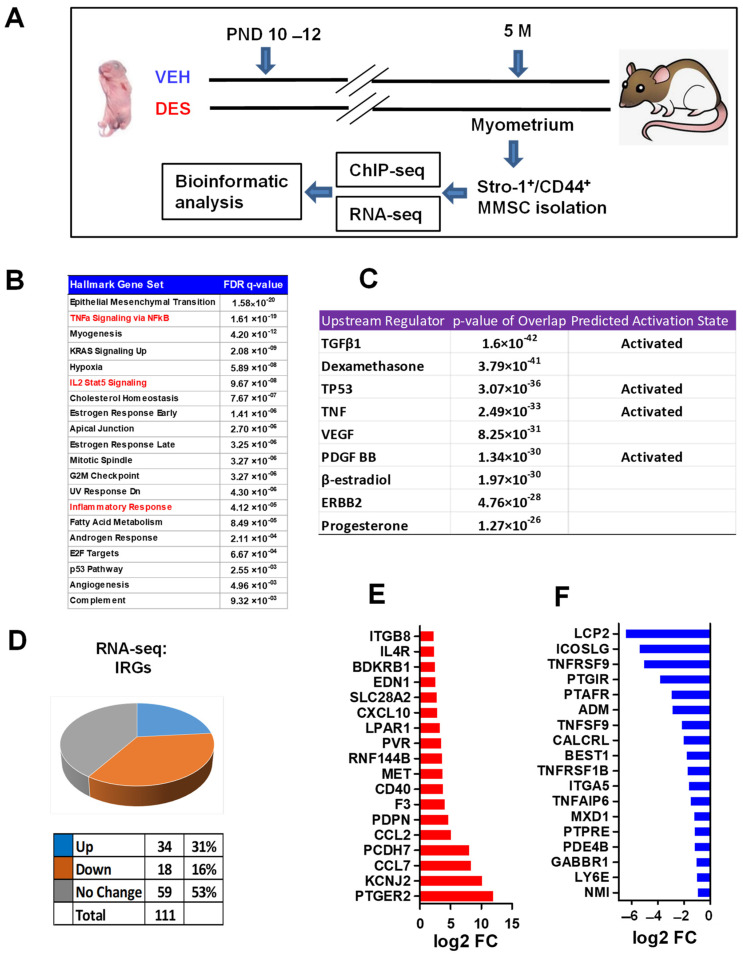
Developmental DES-exposure-activated inflammation pathway in MMSCs. (**A**) Experimental paradigm. Eker rat pups were exposed to VEH and DES at postnatal days 10–12, respectively. The pups were euthanized at five months of age, representing the early adult stage. Myometrial tissues were isolated from the animals and subjected to MMSC isolation using Stro-1/CD44 surface markers. Myometria from five animals were pooled for each treatment. Multi-omics analyses, including RNA-seq and ChIP-seq, were performed to determine the transcriptome and histone modification alterations, respectively. (**B**) Hallmark gene set analysis between DES-MMSCs and VEH-MMSCs. Inflammation-related pathways are highlighted in red. (**C**) Upstream regulator and predicted activation analysis via Ingenuity Pathway Analysis (IPA). (**D**) Pie chart showing the percentage of IRGs that exhibited changes in RNA expression between DES-MMSCs and VEH-MMSCs, as measured via RNA-seq; the cutoff value is 2-fold with an FDR < 0.05. (**E**) List the top 18 IRGs showing differential upregulation in DES- vs. VEH-MMSCs. (**F**) List the top 18 IRGs showing downregulation in DES- vs. VEH-MMSCs.

**Figure 2 ijms-24-11641-f002:**
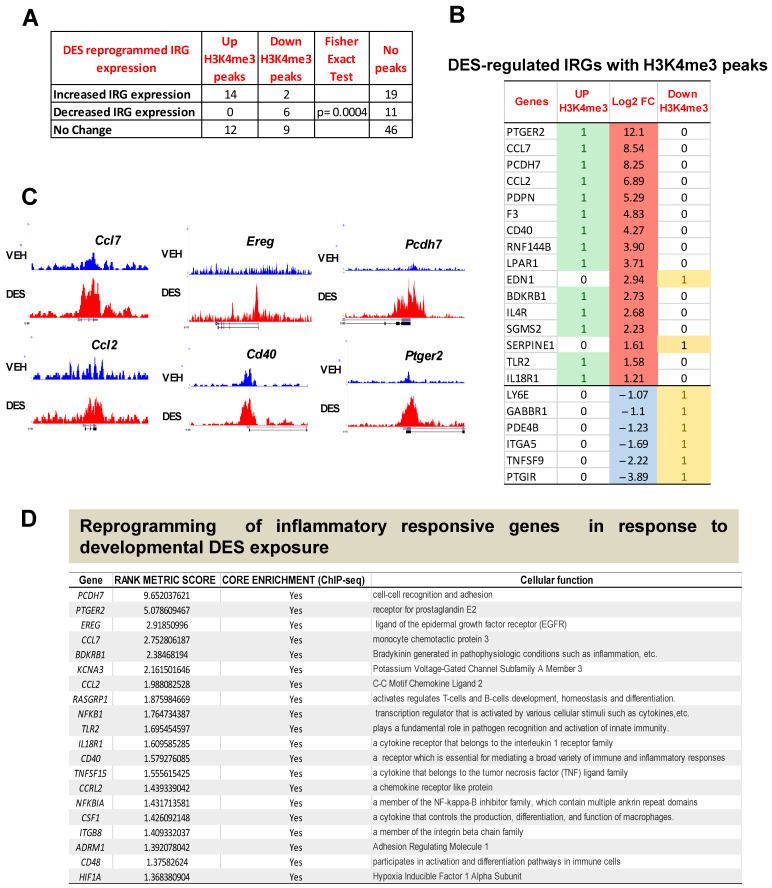
The correlation between RNA expression and H3K4me3-reprogrammed genes. DES-MMSCs and VEH-MMSCs were isolated and subjected to immunoprecipitation with anti-H3K4me3 antibody. ChIP-seq and bioinformatic analysis were performed as described in the Materials and Methods Section. (**A**) DES-regulated IRGs with enrichment or reduction of H3K4me3. (**B**) The list of IRGs with H3K4me3 status. Up-DEGs were highlighted in red. Down-DEGs were highlighted in light blue color. Up-H3K4me3 were highlighted in green color. (**C**) Histograms created using Integrative Genomics Viewer showing H3K4me3 occupancy at *Ccl7*, *Ccl2*, *Ereg*, *Cd40*, *Pcdh7*, and *Ptger2*. For each gene, the upper and lower browser images display an expanded view of a selected region of the H3K4me3 peak distributions in VEH-MMSCs (blue track) and DES-MMSCs (red track). (**D**) The list of top 20 reprogrammed IRGs showing their cellular functions.

**Figure 3 ijms-24-11641-f003:**
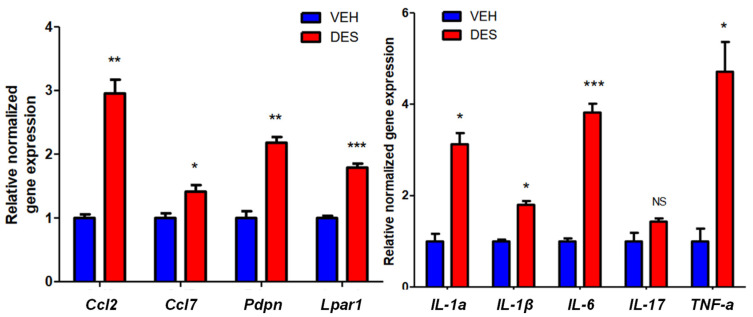
The effect of the secretome of reprogrammed DES-MMSCs on DMC. Serum-free conditioned medium (CM) was prepared from DES-MMSCs and VEH-MMSCs. DMCs from adult rat myometria were grown in the CM from DES-MMSCs and VEH-MMSCs for two days. Bar plots show the relative normalized gene expression of critical inflammation-related genes, including *Ccl2*, *Ccl7*, *Pdpn*, and *Lpar1* (left panel), as well as *IL-1a*, *IL-1b*, *IL-6*, *IL17*, and *TNFa* (right panel). * *p* < 0.05; ** *p* < 0.01, *** *p* < 0.001, NS, no significant difference.

**Figure 4 ijms-24-11641-f004:**
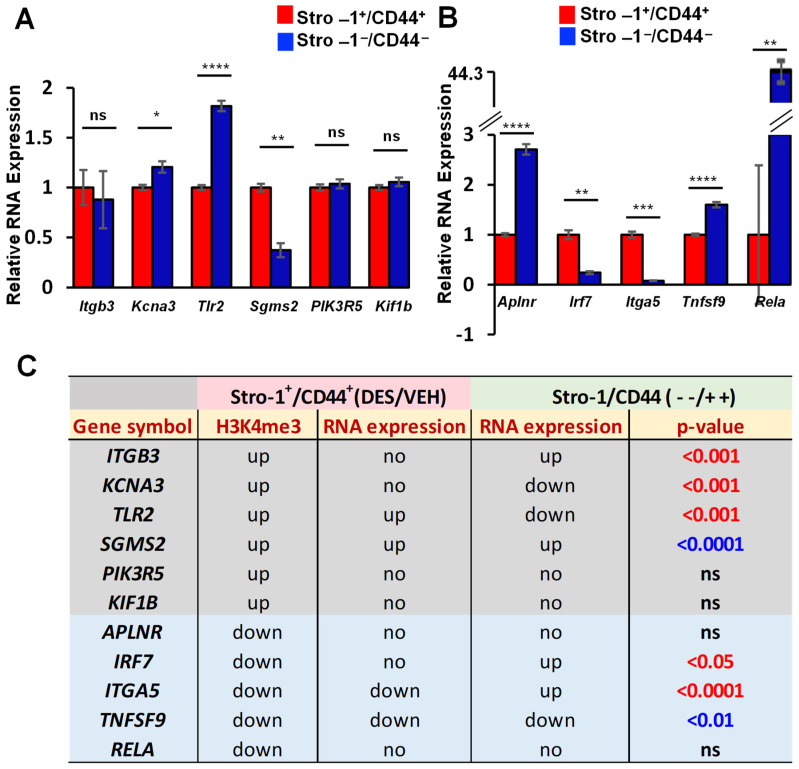
Specific reprogramming of IRGs in MMSCs. (**A**) Bar plots showing the differential expression of IRGs, including *Itgb3*, *Kcna3*, *Tlr2*, *Sgms2*, *Pik3R5*, and *Kif1b*, between DES-MMSCs and DES-DMCs. (**B**) Bar plots showing the differential expression of IRGs, including *Aplnr*, *Lrf7*, *Itga5*, *Tnfsf9*, and *Rela*, between DES-MMSCs and DES-DMCs. (**C**) The correlation between H3K4me3 status and the expression of reprogrammed IRGs (the expression comparison between DES-MMSCs and DES-DMCs, or expression comparison between DES-MMSCs vs VEH-MMSCs). ns: no significant difference. The genes with a grey color background are H3K4me3-enriched genes. The genes with the reduced H3K4me3 are presented with a light blue background. The *p*-value shows the significant difference in gene expression between Stro-1/CD44 double-positive and double-negative cells. Additionally, the *p*-values with the blue color show two comparisons of gene expression (Stro-1/CD44 double-positive vs. double-negative cells, or Stro-1+/CD44+ DES vs. VEH) going in the same direction. The *p*-values with the red color indicated that two comparisons go in the opposite direction, or one comparison shows either up- or down-regulated, and the other shows no significant changes at all. * *p* < 0.05; ** *p* < 0.01, *** *p* < 0.001, **** *p* < 0.0001. ns: no significant difference. no: no significant changes.

**Figure 5 ijms-24-11641-f005:**
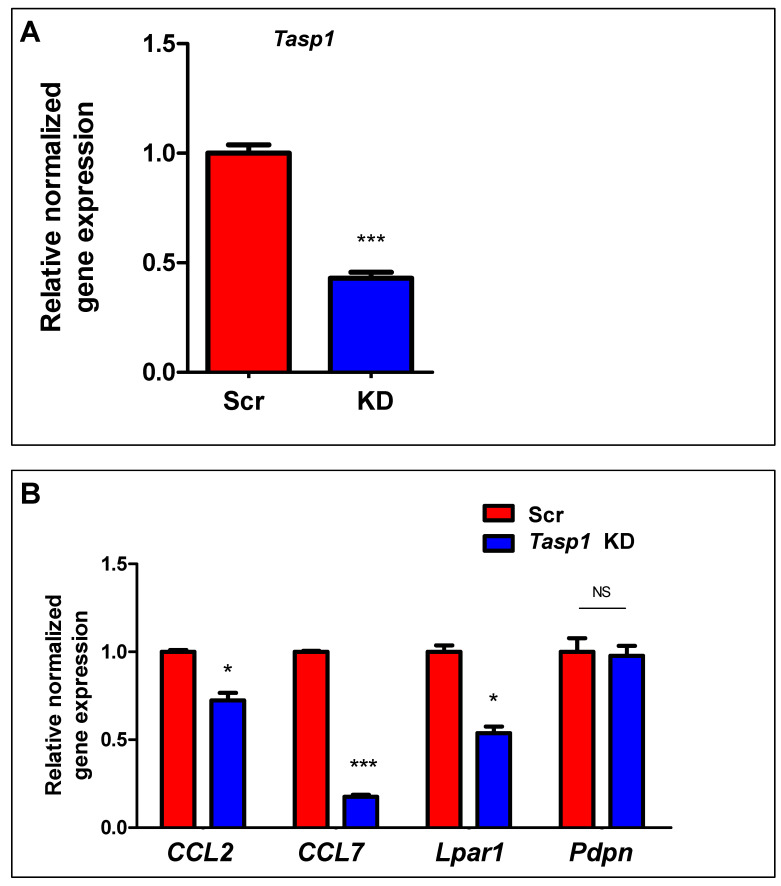
TASP1 is required for the regulation of IRGs expression. (**A**) Bar plots showing the relative normalized gene expression of *Tasp1* RNA levels in DES-MMSCs infected with *Tasp1* rat shRNA lentiviral particles (KD) or scrambled particles (Scr). (**B**) Bar plots showing the relative normalized gene expression of IRGs (*Ccl2*, *Ccl7*, *Lpar1,* and *Pdpn*) after knockdown of *Tasp1.* Scr: scramble; KD: knockdown; * *p* < 0.05; *** *p* < 0.001, NS: no significant difference.

**Figure 6 ijms-24-11641-f006:**
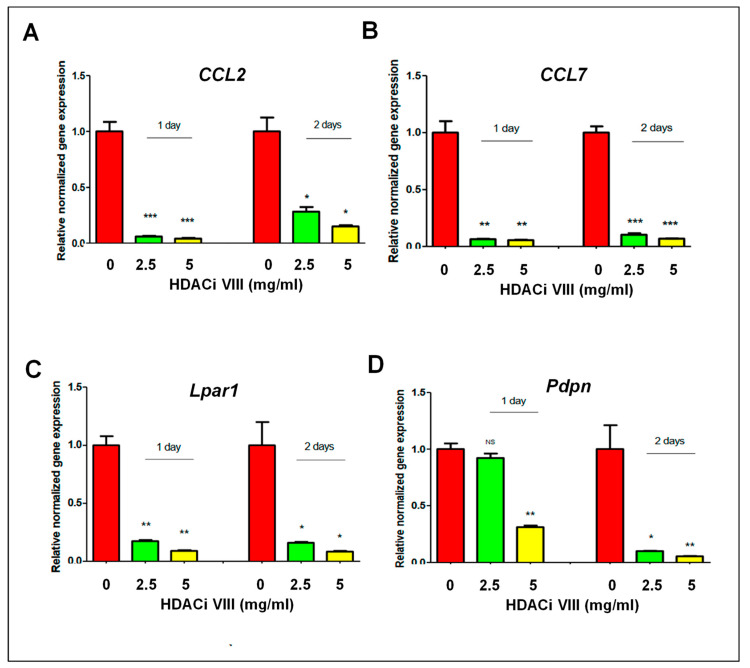
Inhibition of HDAC activity on the expression of reprogrammed IRGs. Bar plots showing the relative normalized gene expressions of (**A**) *Ccl2*, (**B**) *Ccl7*, (**C**) *Lpar1*, and (**D**) *Pdpn* in DES-MMSCs treated with HDAC inhibitor HDACi VIII at concentrations of 2.5 and 5 mg/mL for 1 and 2 days, respectively. * *p* < 0.05; ** *p* < 0.01; *** *p* < 0.001; NS: no significant difference.

**Figure 7 ijms-24-11641-f007:**
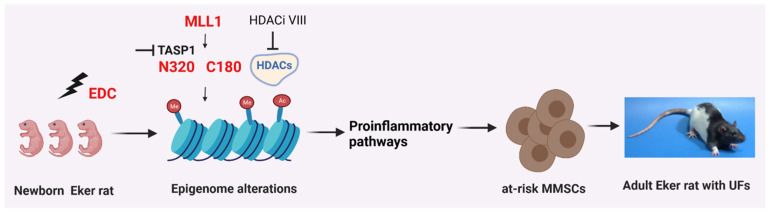
Proposed animal model. Exposure to endocrine-disrupting chemicals (EDCs) during development alters the characteristics of myometrial stem cells (MMSCs) and triggers inflammatory pathways within these cells, which are the source of uterine fibroids (UFs). The abnormal expression of reprogrammed inflammatory responsive genes (IRGs) in MMSCs is driven by the activation of mixed-lineage leukemia protein-1 (MLL1). The EDC, acting as an environmental risk factor, disrupts the epigenetic regulation of MMSCs through modifications to histones, leading to the activation of inflammatory pathways, ultimately contributing to the development of UFs. By inhibiting MLL1 and histone deacetylases (HDACs), the reprogramming of IRGs induced by EDC exposure can be reversed. This figure was created using the BioRender software (BioRender.com).

## Data Availability

The authors declare that data (RNA-seq and ChIP-seq) supporting the findings of this study are available in GEO database under accession number GSE157503.

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
