# Peer review of "Epigenetic Modulation of Inflammatory Pathways in Myometrial Stem Cells and Risk of Uterine Fibroids"

_ijms, 2023, doi:10.3390/ijms241411641_

Round 1

Reviewer 1 Report

Reviewer Comments

The present research article on “Epigenetic modulation of inflammatory pathways in myometrial stem cells and risk of uterine fibroids”, has provided evidence demonstrating that developmental exposure to EDC targets inflammatory pathways in MMSCs, the cell source for UFs. For this, they have performed various studies including the characterization of Myometrial Stro-1+/CD44+ Cells from Eker Rats Exposed to DES and VEH, Whole-transcriptome RNA Sequencing (RNA-seq), and Quantitative Real-Time RT-PCR 99 (RT-qPCR), Chromatin Immunoprecipitation Sequencing (ChIP-seq), HDAC inhibition in MMSCs from Eker Rats and effect of MMSC Secretome on the expression of proinflammatory factors in DMCs. The research study concludes that targeting MLL1 and HDAC epigenetic regulators can reverse the notable changes in the expression of IRGs in MMSCs from Eker rats exposed to EDC.

The research work is well-planned, material methods are discussed briefly, and results and discussed with previous outcomes. The paper is accepted after the incorporation of the following corrections or incorporation of suggestions in the revised MS.

Scientific comments

1.      Line 204, Exposed with both simultaneously? or exposed with any one of the two as mentioned in the method.

2.      Line 42-43, Which environmental exposures are harmful please mention them.

3.      Lines 75-76, provide the specific reference of the method.

4.      Line 185, Either or treated with both.

5.      In Figure 1D, 34 genes are up-regulated however 18 are down-regulated. Why in Figure 1E, you have mentioned only 18 up-regulated genes? What is the meaning of the top 18 here?

6.      As you mentioned in lines 231-232, you have checked 16 genes analyzed and in Figure 2D you have listed the top 20 reprogrammed IRGs showing their cellular functions.

7.      Line 260, What are the basic criteria for the selection of these four Chemokine?

Minor corrections/suggestions

8.      Reference numbers should be placed in square brackets.

9.      Line 102 (DES)-,,....check the typo error.

10.  Line 188, (See the experimental design in Fig.1A)....Repeated sentence. delete it.

11.  line 202, Fig 1. delete it.

12.  Line 243. Delete it.

13.  In Figure 2, bold the font of A, B, C, and D.

14.  Delete the word Fig 3 written in red bold just before Figure 3 legends.

15.  Line 286, elsewhere you have written Fig. follow the same patter.

16.  Line 287, delete it.

17.  Delete the words Fig. 5 from line 313, Fig. 7 from line 468.

18.  In Figure 6 legends, bold (A), (B), and (C).

19.  Check the pattern of reference. The year should be in bold format. Revised accordingly.

Author Response

The present research article on “Epigenetic modulation of inflammatory pathways in myometrial stem cells and risk of uterine fibroids”, has provided evidence demonstrating that developmental exposure to EDC targets inflammatory pathways in MMSCs, the cell source for UFs. For this, they have performed various studies including the characterization of Myometrial Stro-1+/CD44+ Cells from Eker Rats Exposed to DES and VEH, Whole-transcriptome RNA Sequencing (RNA-seq), and Quantitative Real-Time RT-PCR  (RT-qPCR), Chromatin Immunoprecipitation Sequencing (ChIP-seq), HDAC inhibition in MMSCs from Eker Rats and effect of MMSC Secretome on the expression of proinflammatory factors in DMCs. The research study concludes that targeting MLL1 and HDAC epigenetic regulators can reverse the notable changes in the expression of IRGs in MMSCs from Eker rats exposed to EDC.

The research work is well-planned, material methods are discussed briefly, and results and discussed with previous outcomes. The paper is accepted after the incorporation of the following corrections or incorporation of suggestions in the revised MS.

Scientific comments

  1. Line 204, Exposed with both simultaneously? or exposed with any one of the two as mentioned in the method.

Response: Thanks very much for the comment. It has been clarified.

  1. Line 42-43, Which environmental exposures are harmful please mention them.

Response: Thanks very much for the comments. The harmful exposures have been described in the text.

  1. Lines 75-76, provide the specific reference of the method.

Response: The reference has been included.

  1. Line 185, Either or treated with both.

Response: Thanks very much. We have double-checked and confirmed that the description in the text is correct.

  1. In Figure 1D, 34 genes are up-regulated however 18 are down-regulated. Why in Figure 1E, you have mentioned only 18 up-regulated genes? What is the meaning of the top 18 here?

Response: Thanks very much for raising the concern. The top 18 genes were listed based on the fold changes. We listed 18 upregulated genes matching the number of the down-regulated genes. In this manuscript, we focus on the reprogrammed IRGs.

  1. As you mentioned in lines 231-232, you have checked 16 genes analyzed and in Figure 2D you have listed the top 20 reprogrammed IRGs showing their cellular functions.

Response: Thanks very much for raising the concern. We analyzed 16 genes that are both upregulated and reprogrammed by H3K4me3.  For  Fig 2D, we listed 20 out of  26 IRGs with the increase in  H3K4me3 without considering the change of RNA expression.

  1. Line 260, What are the basic criteria for the selection of these four Chemokine?

           Response: Thanks very much for the comments. The rationale for selecting these markers             has been included in the text. In addition, the important role of these chemokines was   described in lines 218-221.

 Minor corrections/suggestions

  1. Reference numbers should be placed in square brackets.

Response: We have made a corresponding change.

  1. Line 102 (DES)-,,....check the typo error.

Response: Thanks very much. The typo error has been corrected.

  1. Line 188, (See the experimental design in Fig.1A)....Repeated sentence. delete it.

Response: Thanks very much. We have deleted the repeated sentence.

  1. line 202, Fig 1. delete it.

Response: Thanks very much. We have deleted it.

  1. Line 243. Delete it.

Response: Thanks very much.

  1. In Figure 2, bold the font of A, B, C, and D.

Response: the font of A, B, C, and D has been bolded.

  1. Delete the word Fig 3 written in red bold just before Figure 3 legends.

Response: Thanks. It has been deleted.

  1. Line 286, elsewhere you have written Fig. follow the same pattern.

Response: Thanks very much. We have checked the figure abbreviation throughout the whole manuscript and have made changes accordingly.

  1. Line 287, delete it.

Response: It has been deleted.

  1. Delete the words Fig. 5 from line 313, Fig. 7 from line 468.

Response: They have been deleted.

  1. In Figure 6 legends, bold (A), (B), and (C).

Response: Thanks. They are bolded.

  1. Check the pattern of reference. The year should be in bold format. Revised accordingly.

 Response: Thanks very much. We have revised the format of references accordingly.

Reviewer 2 Report

Very interesting work on the topic of basic science in uterine fibroid medicine.

It would be worth rewording the conclusions section to make it more interesting for the practicing physician. In my opinion, it is worth proposing possible future practical and clinical benefits for women suffering from uterine myomas based on the results of the evaluated work. 

In my opinion, it fully merits publication after a small correction to the conclusion section summarising the results of the work.

Author Response

Comments and Suggestions for Authors

Very interesting work on the topic of basic science in uterine fibroid medicine.

 It would be worth rewording the conclusions section to make it more interesting for the practicing physician. In my opinion, it is worth proposing possible future practical and clinical benefits for women suffering from uterine myomas based on the results of the evaluated work. 

In my opinion, it fully merits publication after a small correction to the conclusion section summarizing the results of the work.

Response: Thanks very much for the comments.  Per suggestions,  we describe the challenges of understanding EDC -induced uterine diseases and deep dive into the molecular mechanism will help to develop effective prevention and treatment options for UF pathogenesis induced by EDCs.